# Novel potential of low calorie plant burger: Functional turkey meat formulation optimized by replacing quinoa, chia, soy, amaranth and peas as vegetable protein and their influence on texture and sensory traits

Zeinab Erfanian, Marjan Nouri ᴰ*

Department of Food Science and Technology, Ro. C., Islamic Azad University, Roudehen, Iran.

* Marjan.nouri@iau.ac.ir

## Abstract

This study explores the creation of a nutritionally and low-calorie turkey burger by integrating plant-based protein concentrates. Turkey burgers formulated with quinoa, chia, soybean, amaranth and pea proteins at inclusion levels of 0 and 30%. A Taguchi L8 orthogonal array employed to evaluate the effects on water (WHC) and oil (OHC) holding capacities, cooking loss, chemical properties, emulsion activity and stability. Optimal formulations including quinoa, soybean and amaranth at levels of 0, 12.5 and 25% developed using response surface methodology with a central composite design, emphasizing texture and overall acceptability and also microstructure analyzed through scanning electron microscopy (SEM). The glutamic acid identified as the most abundant amino acid across all protein types. The burgers made entirely of turkey meat displayed the lowest pH (6.13) and protein content (17.36%). In contrast, the meat free samples exhibited higher moisture, fat, ash and fiber content along with improved WHC and also OHC and reduced cooking loss. Plant protein formulations showed enhanced elasticity and lower levels of hardness, cohesiveness and chewiness compared to the meat-only samples. Sensory evaluations indicated an inclusion preference for protein concentrates with the optimal formulation consisting of 25% quinoa, 11.86% soybean, and 25% amaranth. SEM analysis confirmed the successful integration of vegetable proteins into the burger matrix. These findings highlighted the potential for mass-producing turkey burgers with reduced meat content, enhanced nutritional value, functional and sensory properties.

## 1. Introduction

Today, many edible products are manufactured and marketed industrially, gaining a special place in shopping baskets for consumers and society due to their appealing

**Data availability statement:** All relevant data are within the manuscript and its Supporting Information files.

**Funding:** The author(s) received no specific funding for this work.

**Competing interests:** The authors have declared that no competing interests exist.

sensory attributes and quick preparation [1]. Turkey meat is popular for high protein, low saturated fat and cholesterol content, which retains volume when it is cooked, unlike red meat [2]. Recently, all products of turkey meat such as sausages, burgers, nuggets, and ham are ready to use in markets [3]. One of the major reasons for criticizing products of meat is saturated fatty acids, which are a risk agent related to colon cancer and coronary heart disease [4]. The functional and biological potentials of vegetable proteins make these foods without fatty acids and cholesterol become products with multiple functions in maintaining nutrient balance for the human body [5].

Quinoa is an important agricultural product that cultivates in South America for a thousand years [6]. The quinoa protein determines from 12 to 23% that globulins (37%) and albumins (35%) are the main components [7]. Chia seeds (*Salvia Hispanica* L.) from the Labiatae plant of the Salvia family have an extremely nutritional value [8]. The European Union has defined approximate component of chia containing 91–96% dry matter, 30–35% fat, 21–45% carbohydrates, 20–22% protein, 18–30% crude fiber, and 4–6% ash [9]. Nut soybean is from pea family and applies in many products containing flour, protein, tofu, milk, sauce, and soybean oil [10]. The soybean is extremely high, but calories are not so much, such as 100 g contains 36 g protein, 30 g carbohydrates, 20 g fat, 9 mg fiber, 155 mg isoflavones, and 160 mg phytosterols [11]. Amaranth is a gluten-free grain that is high in fiber and protein, making it a nutritious choice that provides essential micronutrients including manganese, magnesium, phosphorus and iron [12]. The amaranth protein (about 14%) is higher than other grains with a balanced distribution and high amino acid bioavailability, such as threonine, leucine, valine, and isoleucine [7]. Pea seed is a prominent global commodity and considers a significant source of proteins approximately 25% by weight, which has an amino acid profile comparable to other common legumes [13].

In previous studies, the influence of adding amaranth flour on meat patties [14], quinoa protein emulsions in low-fat frankfurters [15], hemp, amaranth, and flax seeds for chicken patties [12], soybeans, peas, lentils, beans, chlorella and spirulina in turkey burgers [2], barley, potatoes, apples, wheat, and bamboo fibers [16] and also amaranth and pumpkin seeds on chicken [17], quinoa flour [18] and zea [6] in beef burgers studied. The previous studies conducted on optimization of meat flavoring manufacture for plant-based products using the Taguchi method [19], protein fiber spinning via rheological and physicochemical analyses [20] and also a gluten and soy-free enriched with anthocyanins microcapsules [21]. However, adding a combination of quinoa, chia, amaranth, soybean and pea protein concentrates did not investigate in turkey burger formulation.

The study aimed to develop a low-calorie and high-nutrition turkey burger by partially replacing meat with plant-based protein concentrates: quinoa, chia, amaranth, soybean and pea. This research contributed to the development of innovative and healthier meat products by integrating a unique combination for plant-based proteins into a turkey burger. The approach enhanced nutritional value and functional properties, which could help reduce health risks associated with conventional meat products. This research focused on improving a low-calorie product with high nutritional value by replacing

turkey meat with vegetable protein. Initially, amino acid profiles of proteins from quinoa, chia, soybean, amaranth and pea seeds were determined. Following this, eight experiments conducted to assess the impact of five protein variables on various responses including physicochemical functions, water (WHC) and oil (OHC) holding capacities, cooking loss, emulsion stability (ES) and activity (EA). These experiments utilized the Taguchi design and in the subsequent step, three response variables significantly influenced turkey burger production selected for optimization using the central composite design (CCD) within the response surface methodology (RSM) that focused on texture analysis and sensory evaluation.

## 2. Materials and methods

### 2.1. Chemicals

Quinoa, chia, soybean, amaranth, and pea protein concentrates purchased from Wind Mill Co, Netherlands. Phenolphthalein, o-phthalaldehyde, ethanol, hydrochloric acid, and potassium hydroxide prepared from Merck, Germany.

### 2.2. Amino acid content assay of protein concentrate

The non and essential amino acids were assessed using prior digestion with hydrochloric acid 37%, pre-column derivatization o-phthalaldehyde, high-performance liquid chromatography (HPLC, 1260 Infinity, Agilent Technologies, Santa Clara, CA, USA) equipped with a diode array detector (Konik-Tech, Barcelona, Spain) for all amino acids, except tryptophan. An essential digestion, HPLC, and fluorescence detector (Shimadzu RF-10AXL, Kyoto, Japan) performed, and results are stated in g amino acid/ 100 g protein [22].

### 2.3. Preparation of turkey burger samples

Turkey meat sourced from an industrial slaughterhouse along with ingredients such as breadcrumbs, onion, salt and spices purchased from a market in Tehran. The turkey meat, a mix of breast and leg minced using a Kenwood CH580 grinder (Barcelona, Spain). Subsequently, the minced turkey meat and protein concentrates comprising 82% formulation per statistical design blended with other ingredients: onion (15%), salt and spices (2%). Protein concentrates from quinoa, chia, soy, amaranth and pea incorporated at 0 and 30% levels according to the Taguchi design, labeled as seed proteins $SP_1$ to $SP_8$. Next, quinoa, soy, and amaranth measured at 0, 12.5 and 25% per RSM, resulting in plant burgers labeled $PB_1$ to $PB_{20}$. The resulting dough shaped into 70 g molds, frozen and then stored in a refrigerator (Eastcool, Iran) for approximately 12 h before analysis (**Fig 1**).

### 2.4. Physicochemical attributes of burgers

AOAC 2000 method determined moisture, total fat, protein, ash, total dietary fiber, and pH of burgers measured using a portable digital pH meter (Metrohm 827, Switzerland) [23]. Moisture determined based on the weight loss for samples after 12 h of placing them in a 105 °C oven (Shimadzu, CTO-10AS, and Japan). Protein and fat measured based on total nitrogen (N × 6.25) by Keldahl method, and a Soxhlet apparatus, respectively. Total ash for fresh burger samples analyzed by gravimetric procedure and based on burning in the oven. Total, soluble, and insoluble dietary fiber were determined using an enzymatic-gravimetric procedure using kit (Sigma-Aldrich, TDF-100, USA) [12].

### 2.5. WHC and OHC assay measurement

In the present method, each sample about 5 g used, and 30 mL distilled water added, and hydrated samples. Afterwards, mixture centrifuged by centrifuge (Dlab, DM0636, Germany) at 3000 × g for 30 min, supernatant discarded, and a tube with the residue inverted to stand for 5 min [15]. For OHC assay, each sample about 2 g blended with 20 mL corn oil, and tubes were vortexed for 10 min to homogenize. The obtained mixture placed about 30 min at room temperature before centrifugation [24].

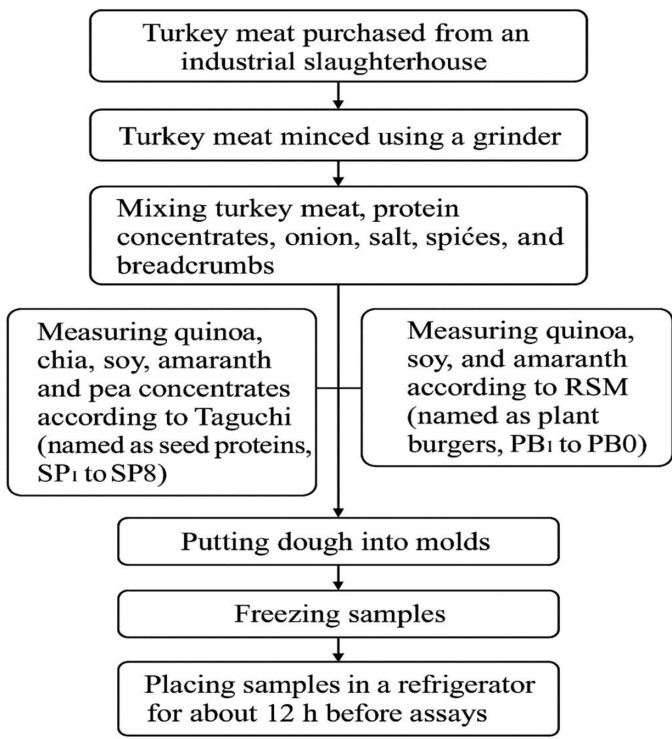

**Fig 1. The obtained flow chart for formulation preparation of turkey burger samples.**

### 2.6. Determination of EA and ES

Each sample about 1 g put into a 50 mL graduated centrifuge tube, 12.5 mL distilled water added, and the sample homogenized with a blender for 1 min. After this procedure, 12.5 mL corn oil with the mixture blended for another 1 min, and emulsion produced in each tube was heated in a water bath at 80 °C for 30 min, cooled to 25 °C, and again centrifuged [25].

### 2.7. Cooking loss

The cooking loss of enriched burgers measured based on the weight difference and conditions were set at 180 °C for 10 min until the meat reached an internal 80 °C. The meat heat determined with a penetration probe; after baking, samples cooled for 30 min at 25 °C. The cooking loss calculated as the weight difference percentage between the mass before and after cooking as following Eq. 1:

$$Cooking\ loss\ (\%) = (W_1 - W_2)\,/W_1 \times 100 \tag{1}$$

Where $W_1$ and $W_2$ represent the weight of uncooked (g) and cooked meat (g), respectively [26].

### 2.8. Tissue analysis

A texture analyzer (TA-XT Plus 100CT, UK) utilized for samples with 80 mm diameter cylindrical probe at speed of 1.0 mm/S. The hardness, cohesiveness, chewiness, elasticity, gumminess, and adhesiveness characteristics were assessed for treated samples [5].

## 2.9. Sensory evaluation

Sensory evaluation conducted with 40-trained panelists (20–40 years old, equal male and female) in ISO 8589-compliant booths. Panelists underwent three 2 h training sessions to familiarize them with turkey burger sensory attributes (appearance, color, smell and elasticity) and the scoring system. Training included reference samples (commercial turkey burgers and plant-based analogs) to calibrate perceptions of texture, flavor and aroma, ensuring consistent evaluation. Samples fried for 4 min, coded with random 3-digit numbers and presented in a randomized order to prevent sequence bias. Single-blinding ensured panelists were unaware of sample formulations and attributes were scored on a 5-point hedonic scale: 5 (excellent), 4 (good), 3 (acceptable), 2 (poor), 1 (unacceptable). The scale was calibrated during training to minimize subjectivity with "poor" indicating noticeable defects (e.g., off-odor and gritty texture) and "unacceptable" exhibiting rejection due to severe flaws (e.g., rancid flavor and disintegrating texture). Panelists cleaned plates with water and unsalted crackers served between samples to avoid carryover effects [6].

## 2.10. Morphology assay by scanning electron microscopy (SEM)

The SEM (Tescan, Model Mira3, and Czech) evaluated morphology surface, and cross section of control and optimal samples coated with a thin gold layer by a direct current sputtering method. Microscopy assessment (× 250) was conducted in secondary electron mode with 15 kV accelerating voltage at 17 mm of effective working distance [17].

## 2.11. Statistical analysis

**2.11.1. Taguchi design.** The Taguchi method screened the effects of five plant-based protein concentrates (quinoa, chia, soybean, amaranth and pea) at two levels (0 and 30%) on physicochemical attributes, WHC, OHC, ES, EA and cooking loss using an L8 orthogonal array (8 runs, $SP_1$ to $SP_8$). The signal-to-noise ratio used "smaller-the-better" for cooking loss and "larger-the-better" in WHC, OHC, EA and ES. Normality (Shapiro-Wilk) and homogeneity of variances (Levene test) verified and also ANOVA assessed significance ($p < 0.05$) with post hoc test comparing means for Tukey. Quinoa, soybean, and amaranth selected for RSM based on significant effects and also model validation confirmed predicted vs. observed values (< 5% deviation).

**2.11.2. RSM design optimization.** RSM with a CCD optimized quinoa, soybean and amaranth (0, 12.5 and 25%) for texture and sensory acceptability across 20 runs ($PB_1$ to $PB_{20}$). A second-order polynomial model fitted with normality, homogeneity and independence verified (Shapiro-Wilk, Levene and Durbin-Watson tests). ANOVA confirmed significance ($p < 0.05$, $R^2 > 0.90$) with non-significant lack of fit ($p > 0.05$). Tukey test compared means and validation that showed < 10% deviation between predicted and observed responses.

## 3. Result and discussion

### 3.1. Amino acid profile of protein concentrates

The amino acid profiles of quinoa, chia, soybean, amaranth and pea protein concentrates revealed glutamic acid as the most abundant across all sources (Table 1, Fig 2). Amaranth had the highest essential amino acid content (e.g., leucine, threonine, valine), followed by quinoa, chia, soybean and pea. Essential amino acids are vital for human nutrition, as they synthesized endogenously and must be dietary-sourced. Amaranth elevated leucine (6–8 g/ 100 g protein) supported muscle protein synthesis, critical for growth, repair and maintenance, particularly in active individuals or the elderly. Threonine, abundant in amaranth and quinoa (~ 4 g/ 100 g), aids immune function and mucin production for gut health. Lysine with higher level in pea and soybean (~6 g/ 100 g) complements cereal-based diets and addressing deficiencies in populations reliant on grains. These differences are nutritionally significant as plant-based proteins with balanced essential amino acid profiles can reduce dependence on animal proteins, supporting vegetarian, flexitarian or deficient diets.

**Table 1. The component results of amino acid profiles in quinoa, chia, soy, amaranth and pea protein concentrates.**

| | Amino acid | Proteins (g/100 g) | | | | |
|---|---|---|---|---|---|---|
| | | Quinoa | Chia | Soybean | Amaranth | Pea |
| **Essential amino acid** | Histidine | 7.13±0.1ᵃ | 3.76±0.1ᶜ | 2.51±0.3ᵈ | 5.95±0.1ᵇ | 2.34±0.1ᵈ |
| | Threonine | 6.58±0.1ᵃ | 5.23±0.1ᶜ | 3.78±0.2ᵇ | 3.80±0.2ᵇ | 3.65±0.1ᵇ |
| | Valine | 4.67±0.1ᵇᶜ | 4.92±0.1ᵃ | 4.12±0.1ᵈ | 4.82±0.1ᵃᵇ | 4.56±0.1ᶜ |
| | Lysine | 10.24±0.5ᵃ | 9.31±0.4ᵇ | 8.76±0.3ᶜ | 10.45±0.3ᵃ | 7.12±0.5ᵈ |
| | Isoleucine | 4.09±0.1ᶜ | 2.05±0.1ᵉ | 5.25±0.1ᵃ | 3.53±0.1ᵈ | 5.03±0.1ᵇ |
| | leucine | 5.44±0.2ᵇᶜ | 4.31±0.2ᵈ | 5.82±0.3ᵇ | 5.23±0.4ᶜ | 6.80±0.4ᵃ |
| | Phenylalanine | 4.21±0.2ᵇ | 5.48±0.2ᵃ | 5.31±0.3ᵃ | 4.07±0.2ᵇ | 5.86±0.4ᵃ |
| | Methionine | 1.39±0.1ᶜ | 3.78±0.1ᵃ | 1.54±0.1ᶜ | 1.78±0.1ᵇ | 1.11±0.1ᵈ |
| | Tryptophan | 0.93±0.1ᶜ | 1.45±0.1ᵇ | 1.13±0.1ᶜ | 3.23±0.1ᵃ | 1.15±0.1ᶜ |
| **Total** | | 40.78 | 40.29 | 38.22 | 42.86 | 37.62 |
| **Nonessential amino acid** | Alanine | 3.23±0.1ᵈ | 3.65±0.1ᶜ | 4.79±0.1ᵇ | 5.36±0.1ᵃ | 4.59±0.1ᵇ |
| | Apartic acid | 10.09±0.4ᵃ | 10.56±0.4ᵃ | 10.48±0.3ᵃ | 9.18±0.4ᵇ | 10.95±0.6ᵃ |
| | Glutamic acid | 13.47±0.4ᵇ | 15.62±0.6ᵃ | 15.32±0.6ᵃ | 11.23±0.5ᵈ | 12.42±0.5ᶜ |
| | Serine | 3.78±0.2ᵈ | 6.31±0.4ᵇ | 5.15±0.4ᶜ | 7.58±0.5ᵃ | 5.21±0.5ᶜ |
| | Arginine | 9.65±0.5ᵃᵇ | 10.32±0.6ᵃ | 7.54±0.3ᵈ | 9.32±0.5ᵇ | 8.51±0.4ᶜ |
| | Glycine | 4.59±0.1ᵇ | 4.76±0.3ᵇ | 4.23±0.3ᵇ | 5.12±0.1ᵃ | 3.89±0.2ᶜ |
| | Tyrosine | 1.87±0.1ᵉ | 2.98±0.1ᵈ | 3.76±0.2ᶜ | 4.65±0.2ᵃ | 4.11±0.3ᵇ |
| | Cysteine | 1.89±0.1ᵇ | 1.06±0.1ᶜ | 1.73±0.1ᵇ | 2.12±0.1ᵃ | 0.90±0.1ᶜ |
| **Total** | | 48.50 | 55.26 | 53.09 | 53.56 | 50.58 |
| **Hydrophobic amino acid** | | 24.90 | 27.17 | 30.59 | 29.44 | 32.06 |
| **Aromatic amino acid** | | 13.74 | 13.67 | 12.71 | 17.90 | 13.46 |

Letters a-d indicate significant differences between distinct treatments (each row).

In previous studies, a further level of essential amino acid had been reported 41.07% [1,8]. Hydrophobic amino acids (e.g., leucine and isoleucine) and aromatic amino acids (e.g., phenylalanine and tyrosine), the most abundant in amaranth and quinoa enhance antioxidant activity by interacting with lipid bilayers and scavenging free radicals. This reduces oxidative stress linked to chronic diseases like cardiovascular disease and cancer offering long-term health benefits [27]. Previous studies reported that more hydrophobic amino acids in protein hydrolysates with 39.64% values were one of the important factors on high antioxidant capacity of peptides [4,28]. The balanced amino acid is one of the major quinoa characteristics according to obtained results, and several studies confirm more quality and complete protein with providing varieties [18,27,28]. Similar to the present results, glutamic (11.84 to 15.21 g/ 100 g protein) had been reported as the dominant amino acid in quinoa [22]. Chia protein has a suitable digestibility (78.9%), similar to casein (88.6%), beans (77.5%) and more than corn (66.6%), rice protein (4.59%), and also wheat (52.7%), but it is lower (90%) than amaranth [29]. Soy identifies suitable essential amino acids, and in past studies, non-components such as glutamic had been reported in the highest amount [2,11]. The most and least amino acids were glutamic and methionine in pseudo cereals such as amaranth, respectively [22].

### 3.2. Results related to the evaluation of burger samples through Taguchi design

**Chemical attributes.** The pH level (6.13) and total protein content (17.36%) were lower in SP₁, which contained 82% turkey meat compared to the other samples. Conversely, pH improved with addition of protein, resulting in SP₄ having the

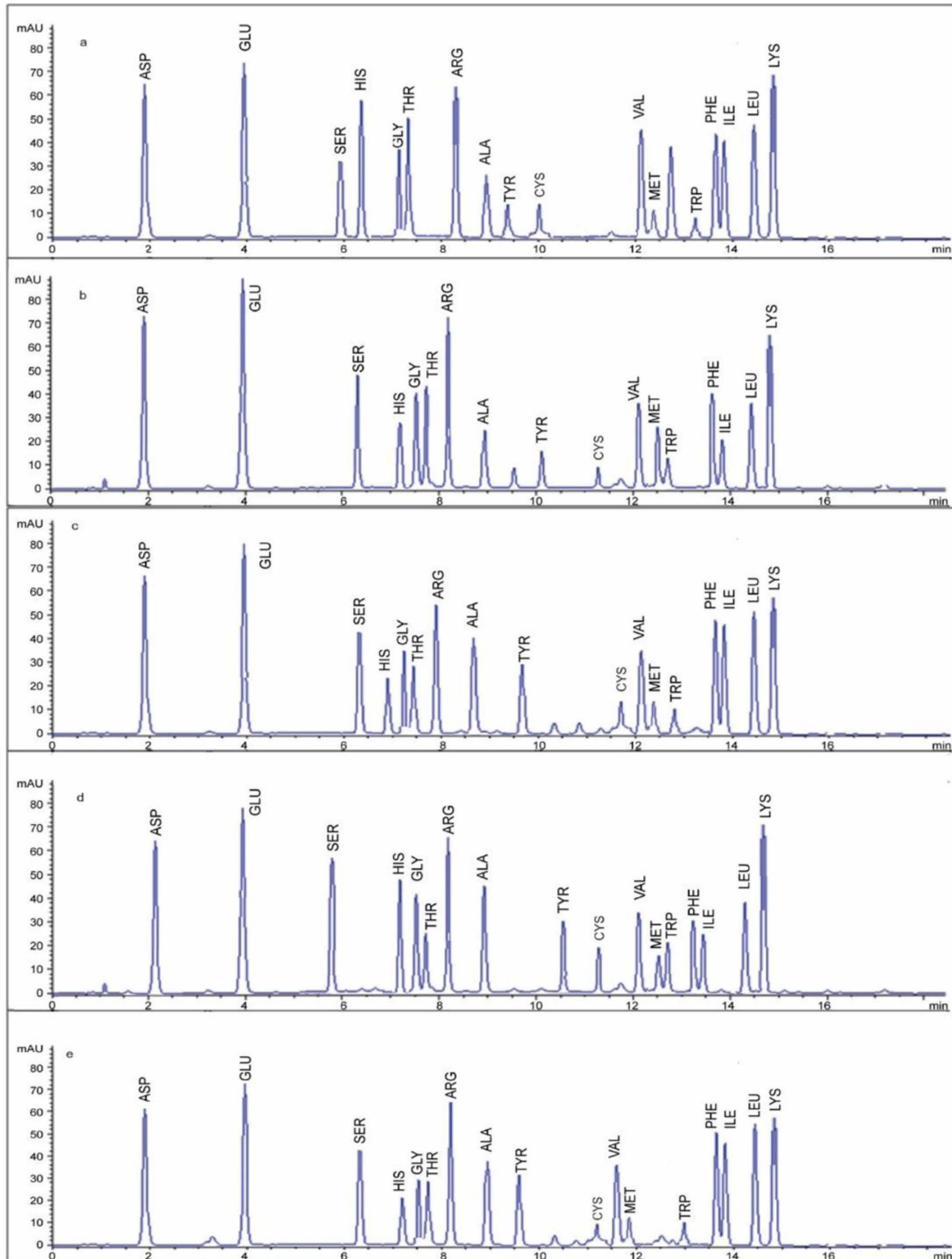

**Fig 2. The high-performance liquid chromatography in quinoa, chia, soy, amaranth, and pea protein concentrates.**

highest values 6.47 and 80.13% for protein content. $SP_1$ exhibited the highest levels of moisture, fat, ash and fiber, while $SP_4$ showed the lowest without turkey meat (**Table 2**).

In all treated samples, the pH for poultry meat remained within the acceptable range of 5.7 to 6.4 [3]. According to meat type and gender, pH value was 5.95 (male breast) to 6.33 (female thigh) and, moisture, protein, and ash reported as 74.7 to 74.8, 20.1 to 24, and 1.15 to 1.27%, respectively [30]. The reported values for turkey indicated 20.5% protein and 1.6% fat [2] and additionally, in a pâté containing 70% meat, the composition included 6.42 pH, 61% moisture, 19.56% protein, 7.90% fat, 1.51% ash and 1.08% fiber. The addition of hemp, amaranth and flax had no significant effect on pH, but enhanced fat, ash, protein and fiber [12].

### 3.2. WHC, OHC, EA and ES evaluation

WHC and OHC are practical features in burger production, and also **Table 2** illustrates that WHC detect from 1.23 ($SP_1$) to 5.4 ($SP_4$) g/g. $SP_4$, $SP_8$, $SP_6$, $SP_7$, $SP_5$, $SP_3$, $SP_2$ and $SP_1$ had the highest WHC and OHC varied from 1.01 ($SP_1$) to 4.95 g/g ($SP_4$), respectively. $SP_4$, $SP_5$, $SP_7$, $SP_6$, $SP_8$, $SP_2$, $SP_3$ and $SP_1$ had the highest OHC, which samples indicated significant differences ($p<0.05$). Since in $SP_2$ and $SP_3$, protein concentrates had replaced 30% turkey meat; therefore, hydrophobic amino acids of amaranth and peas were higher than soy and chia; as a result, WHC level for $SP_3$ was more than $SP_2$; however, the reversed trend observed in OHC. The maximum WHC and OHC obtained for $SP_8$ and $SP_5$ that contained lower absorbent amino acids with more water and the least found in $SP_8$, respectively. According to results from fatty acid profile, pea, soybean, amaranth, quinoa, and chia as protein concentrates had the most hydrophobic amino acids. WHC and OHC are related to availability of polar and nonpolar amino acid residues and also the micro- and macrostructures for protein [15]. The more hydrophilic and polar amino acids on the protein molecule surface help to enhance

**Table 2. Taguchi design and response variables related to the chemical characteristics, water and oil holding capacities, emulsion stability and activity with cooking loss of quinoa, chia, soybean, amaranth, and pea protein concentrates.**

| | Treatments | | | | | | | |
| --- | --- | --- | --- | --- | --- | --- | --- | --- |
| | $SP_1$ | $SP_2$ | $SP_3$ | $SP_4$ | $SP_5$ | $SP_6$ | $SP_7$ | $SP_8$ |
| | Variables (Protein concentrate %) | | | | | | | |
| Quinoa | 0 | 0 | 0 | 0 | 25 | 25 | 25 | 25 |
| Chia | 0 | 0 | 25 | 25 | 0 | 0 | 25 | 25 |
| Soybean | 0 | 0 | 25 | 25 | 25 | 25 | 0 | 0 |
| Amaranth | 0 | 25 | 0 | 25 | 0 | 25 | 0 | 25 |
| Pea | 0 | 25 | 0 | 25 | 25 | 0 | 25 | 0 |
| | Responses | | | | | | | |
| pH | 6.13±0.05[c] | 6.35±0.05[b] | 6.31±0.05[b] | 6.47±0.04[a] | 6.38±0.05[b] | 6.37±0.05[b] | 6.35±0.05[b] | 6.38±0.05[b] |
| Moisture (%) | 62.24±1.57[a] | 30.25±1.05[b] | 30.95±2.05[b] | 10.14±1.05[d] | 22.31±2.5[c] | 22.14±2.6[c] | 22.16±1.5[c] | 22.45±2.8[c] |
| Protein (%) | 17.36±0.25[d] | 57.81±0.83[c] | 57.41±0.61[c] | 80.13±0.93[a] | 64.34±0.68[b] | 64.05±0.49[b] | 64.47±0.55[b] | 64.71±0.80[b] |
| Fat (%) | 12.80±0.05[a] | 6.40±0.05[b] | 6.38±0.05[b] | 0±0.00[d] | 3.23±0.05[c] | 3.18±0.08[c] | 3.10±0.08[c] | 3.25±0.0.08[c] |
| Ash (%) | 0.96±0.07[a] | 0.57±0.07[b] | 0.54±0.06[b] | 0.13±0.02[d] | 0.33±0.05[c] | 0.35±0.03[c] | 0.37±0.05[c] | 0.32±0.03[c] |
| Fiber (%) | 1.04±0.07[a] | 0.78±0.05[b] | 0.76±0.07[b] | 0.18±0.05[d] | 0.25±0.02[c] | 0.27±0.03[c] | 0.25±0.02[c] | 0.23±0.02[c] |
| WHC (g/g) | 1.23±0.07[f] | 1.51±0.15[e] | 1.74±0.23[e] | 5.40±0.20[a] | 2.89±0.20[d] | 4.23±0.13[b] | 3.78±0.30[c] | 4.51±0.50[b] |
| OHC (g/g) | 1.01±0.25[h] | 2.50±0.43[f] | 1.78±0.21[g] | 4.95±0.33[a] | 4.30±0.30[b] | 3.41±0.12[d] | 3.73±0.10[c] | 3.01±0.20[e] |
| Emulsion activity | 14.11±0.15[h] | 26.40±0.15[f] | 24.38±0.15[g] | 41.08±0.15[a] | 40.39±0.18[b] | 33.10±0.15[d] | 39.01±0.20[c] | 32.60±0.20[e] |
| Emulsion stability | 16.08±0.15[h] | 28.57±0.17[f] | 26.54±0.16[g] | 44.38±0.15[a] | 43.32±0.13[b] | 34.12±0.15[d] | 40.02±0.13[c] | 33.52±0.15[e] |
| Cooking loss (%) | 20.14±0.15[a] | 18.23±0.20[b] | 17.94±0.12[c] | 11.45±0.10[f] | 12.86±0.20[d] | 12.02±0.25[e] | 12.54±0.32[de] | 12.13±0.18[e] |

Letters a-g indicate significant differences between distinct treatments (each row).

WHC, while greater availability of hydrophobic residues is responsible for further OHC [26]. Therefore, higher protein cannot contribute to a significant hydrophobic amino acids; however, is also potentially responsible for a greater extent of denaturation, thereby introducing more hydrophobic sites that help to more oil absorption capacity [24]. In the burger review, the WHC levels were 87 and 94% for the meat samples containing vegetable proteins [15]. WHC varied from 1.5 to 2.9 (g/g) in burgers containing pea and soy protein, which was dependent on protein types, but potentially related to available polar amino acids in each sample [24]. In frankfurter sausage, the highest WHC level (84%) obtained for the sample containing 100 quinoa protein [15]. OHC derived from pea protein (0.82 to 1.04 g/g) was significantly higher than those with wheat gluten (0.75 to 0.86 g/g) or soy (69. 0 to 0.84 g/g), which was probably due to more hydrophobic amino acids compared to others [24].

The results for EA and ES show in **Table 2**; therefore, the ranges are 14.11 to 41.08% and 16.08 to 44.38% in $SP_1$ and $SP_4$, respectively.

The critical factor of ES and EA considers amino acids in proteins, which is essential for hydrophobic components on the surface, maintaining hydrophilicity and hydrophobicity balances [26]. In the current research, pea, soybean, amaranth, chia and quinoa exhibited the highest levels of water repellency and $SP_4$ demonstrated specific proteins, resulting in more EA and also ES. In contrast, $SP_1$ that did not contain protein concentrate illustrated the lowest values for these factors, highlighting the significance of composition in burgers. Notably, the emulsifying stability for amaranth (47%) and quinoa (49.2%) was comparable to corn starch (48.8%), which served as control [26]. The mentioned factor for quinoa reported up to 45% [31] and amaranth 7% [25].

The percentage of burger cooking loss in $SP_1$ (20.14%) is significantly ($p < 0.05$) higher than other samples, which are without turkey meat and indicate the lowest (11.45%) level (**Table 2**). $SP_5$, $SP_6$, $SP_7$, and $SP_8$ (12.02 to 12.86%) and also $SP_2$ and $SP_3$ (17.94 to 18.23%) did not illustrate any significant difference.

The cooking loss was measured by determining their changes of weight before and after the procedure [26]. The cooking loss of samples containing turkey meat alone was much more than including vegetable proteins. Cooking loss indicated that liquid loss from turkey burgers was higher than vegetable protein possibly due to poor WHC, as discussed in the previous section. As previously mentioned, the protein in meat products has a spiral transition during heating; as a result, the tissue was soft, and muscle fibers shrank in length and also width and denatured [5]. Because of this procedure, they contracted, and some liquid trapped in them squeezed out; in contrast, structure in plant product did not collapse because the proteins denatured before heating, so a less change seen in overall framework and fluid retention [26]. The lower moisture and cooking loss in the reduced-meat burgers could be because these burgers mainly contained more protein, less moisture, and oil, which helped to retain water in texture of burgers [5]. The present finding was consistent with previous research that burgers containing textured vegetable protein had lower cooking loss than 100% meat with a range of 11–38% after 10 min procedure [26]. In another study, the higher cooking loss in meat patties during cooking attributed to loss of ES for hydrophobic interaction between lipids and water [32]. According to analysis of patty sample, cooking loss for control containing meat was 20.20%, and in samples without meat 12.39 to 15.04% [33]. This factor for sausage samples containing amaranth, quinoa, and corn flour reported as 29, 21 and 33% respectively [26].

### 3.3. Diagrams obtained in the Taguchi scheme

In **Fig 3**, Taguchi's results (pH, protein, moisture, fat, ash, fiber, WHC, OHC, ES, EA, and cooking loss) regarding the main effects of different factors (0 and 30% from quinoa, chia, soy, amaranth, and pea protein concentrates) exhibit, which charts have specified the most favorable conditions for achieving best formulation. Taguchi results showed that quinoa, soybean and amaranth protein concentrates selected for the final burger formulation.

The findings were consistent with existing literature on plant-based protein concentrates in meat products. For example, quinoa protein concentrates enhanced the texture and water retention of beef patties due to their high WHC and gel-forming properties [34]. Similarly, soybean protein concentrates extensively studied for their emulsifying and

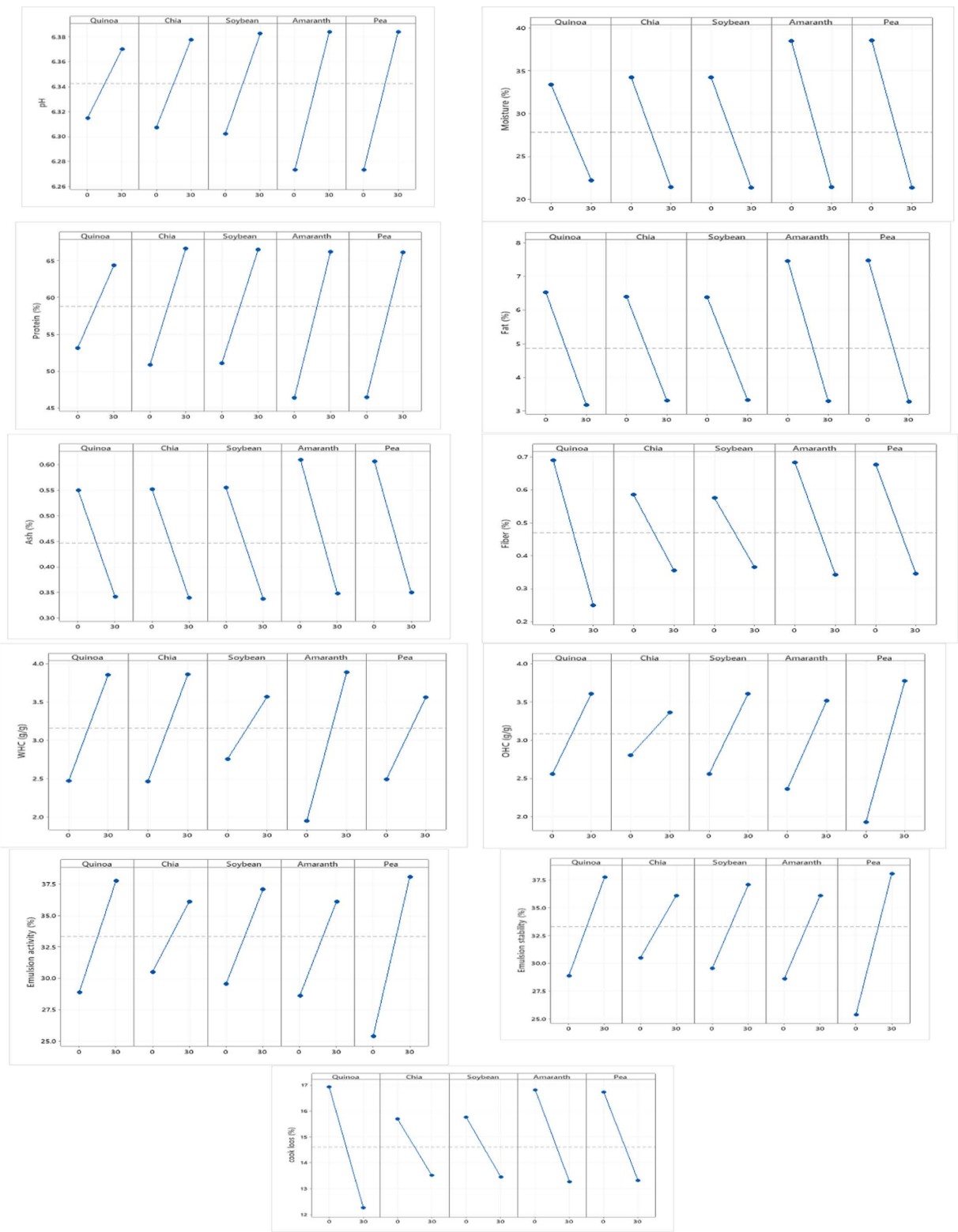

**Fig 3. The obtained diagrams of the Taguchi scheme design in quinoa, soybean, and amaranth protein concentrates.**

fat-binding abilities, making them a common ingredient in meat analog formulations [35]. While less common, amaranth has garnered attention for balanced amino acid profile and mineral content, which improves the nutritional quality of meat products [36]. Combining plant proteins like quinoa and soy can address individual drawbacks such as lower emulsifying capacity and beany flavor resulting in a more balanced formulation, respectively [37]. The Taguchi results in **Fig 2** support this strategy, suggesting that the selected concentrates likely complement each other to optimize multiple response variables. Additionally, using 30% concentration was consistent with research indicating that higher levels of plant protein (20–30%) enhanced functionality without negatively affecting sensory attributes [38]. However, concentrations exceeding 30% could introduce off-flavors or texture problems, which justified the Taguchi scheme on 0 and 30% levels [39].

### Results for RSM.

**The CCD design related to texture**  The results in **Table 3** show that the highest hardness recorded for $PB_1$ at $19.72 \pm 0.09\,N$, while the lowest in $PB_8$ at $9.87 \pm 0.25\,N$. The predicted values from the experiments closely aligned, indicating a good fit for the model. This equation was derived after excluding the variables (A: quinoa, B: soybean and C: amaranth) that did not significantly influence the model:

Hardness (N) = $13.28 - 2.08A - 1.49B - 1.69C + 0.58AC + 0.60BC - 0.52A^2 + 0.99B^2$

ANOVA applied and the results summarized in **Table 4** and also p-value for the model was less than 0.0001, indicating a more significance. The $R^2$ (0.98) and the adjusted $R^2$ (0.97) confirmed the strong significance of model. The analysis demonstrated that variables in linear form (A, B and C), the quadratic terms (A and B) and the interaction effects (AC and BC) all significantly affected the hardness value.

The effect of A and C decreases hardness for samples with improving variables, as depicted in **Fig 4a**. The hardness value declines by influencing BC that, higher C and B up to 12.5%, and after this limit, the reduction process is insignificant in **Fig 4b**.

Based on results for cohesiveness (Table 3), the lowest level obtained for $PB_8$ (0.255) and the highest in $PB_1$ (0.351), and a significant effect not observed on the model by the following equation after removing variables:

Cohesiveness = $0.29 - 0.02A - 0.016B - 0.017C - 0.006AB + 0.01B^2$

According to the ANOVA table, the p-value of the model was less than 0.0001, which indicated $R^2$ (0.95) and $R^2_{adj}$ (0.90) were highly significant. The variables A, B, and C for linear, B in the quadratic model, and AB interaction had a considerable influence ($p < 0.05$) on cohesiveness. The three-dimensional curve of AB interaction indicates in **Fig 4c**, and with increasing variables simultaneously, cohesiveness factor reduces for samples.

According to **Table 3**, the maximum elasticity obtained in $PB_1$ (0.521 mm) and the minimum value for $PB_8$ (0.623 mm). The predicted levels from assays are completely near to each other, and the following equation measured after removing variables that had no considerable influence on the model:

Elasticity (mm) = $0.57 + 0.014A + 0.016B + 0.013C + 0.008AC$

Based on ANOVA, the p value determined in the model was lower than 0.0001, which $R^2$ (0.96) and $R^2_{adj}$ (0.93) proved that model was remarkable. The A, B, and C variables in linear mode and AC interaction exhibited a considerable influence ($p < 0.05$) on elasticity of burgers. According to contour plot illustrating AC interaction (**Fig 4d**), A variable kept constant and elasticity enhanced by higher C.

The gumminess results for the turkey burger samples (**Table 3**) showed that $PB_{10}$ had the lowest gumminess at 2.35 N, while $PB_1$ indicated the highest at 3.36 N. Variables that did not significantly influence the model were measured after being removed using the following equation:

Gumminess (N) = $2.61 - 0.17A - 0.12B - 0.14C + 0.08AC + 0.07BC + 0.13B^2$

$R^2$ (0.97) and adjusted $R^2$ (0.94) indicated that model was highly significant, as determined by ANOVA (**Table 4**). B and C in linear, B for quadratic mode and also the interactions between AC and BC had a significant impact on gumminess in burgers ($p < 0.05$). The gumminess samples decreased with adding variables simultaneously in $p < 0.05$. As illustrated in

Table 3. Central composite design of texture responses (Hardness, cohesiveness, elasticity, gumminess, chewiness and adhesiveness) based on independent variables.

| Run | Independent variables (%) | | | Hardness (N) | | Cohesiveness | | Elasticity (mm) | | Gumminess (%) | | Chewiness (N) | | Adhesiveness (g.s) | |
|---|---|---|---|---|---|---|---|---|---|---|---|---|---|---|---|
| | Quinoa | Soybean | Amaranth | Actual value | Predicted value | Actual value | Predicted value | Actual value | Predicted value | Actual value | Predicted value | Actual value | Predicted value | Actual value | Predicted value |
| PB$_1$ | 0 | 0 | 0 | 19.72±0.09 | 20.013 | 0.351±0.005 | 0.355 | 0.521±0.001 | 0.527 | 3.40±0.01 | 3.105 | 3.36±0.01 | 3.321 | −4.52±0.01 | −4.499 |
| PB$_2$ | 25 | 0 | 0 | 15.39±0.13 | 15.191 | 0.321±0.015 | 0.319 | 0.537±0.003 | 0.536 | 4.25±0.03 | 4.463 | 2.78±0.03 | 2.790 | −5.42±0.03 | −5.340 |
| PB$_3$ | 0 | 25 | 0 | 16.46±0.10 | 16.325 | 0.333±0.006 | 0.332 | 0.556±0.002 | 0.547 | 4.15±0.02 | 4.379 | 2.89±0.02 | 2.898 | −4.98±0.02 | −4.958 |
| PB$_4$ | 25 | 25 | 0 | 10.55±0.10 | 10.493 | 0.275±0.001 | 0.273 | 0.563±0.001 | 0.563 | 4.60±0.01 | 4.487 | 2.46±0.01 | 2.412 | −6.54±0.01 | −6.739 |
| PB$_5$ | 0 | 0 | 25 | 14.28±0.13 | 14.259 | 0.311±0.005 | 0.311 | 0.531±0.002 | 0.528 | 4.10±0.02 | 4.267 | 2.69±0.02 | 2.722 | −5.68±0.02 | −5.477 |
| PB$_6$ | 25 | 0 | 25 | 11.71±0.10 | 11.767 | 0.292±0.180 | 0.291 | 0.564±0.003 | 0.570 | 4.95±0.01 | 4.775 | 2.54±0.01 | 2.516 | −6.25±0.02 | −6.268 |
| PB$_7$ | 0 | 25 | 25 | 12.88±0.25 | 13.001 | 0.293±0.005 | 0.293 | 0.568±0.003 | 0.566 | 4.75±0.02 | 4.591 | 2.63±0.02 | 2.604 | −6.03±0.03 | −6.106 |
| PB$_8$ | 25 | 25 | 25 | 9.87±0.11 | 9.499 | 0.255±0.015 | 0.249 | 0.623±0.001 | 0.614 | 3.50±0.01 | 3.849 | 2.42±0.01 | 2.443 | −7.82±0.01 | −7.837 |
| PB$_9$ | 0 | 12.5 | 12.5 | 15.1±0.09 | 14.839 | 0.315±0.005 | 0.310 | 0.547±0.001 | 0.551 | 4.10±0.01 | 4.157 | 2.71±0.01 | 2.733 | −5.53±0.01 | −5.697 |
| PB$_{10}$ | 25 | 12.5 | 12.5 | 10.11±0.10 | 10.677 | 0.261±0.005 | 0.270 | 0.579±0.002 | 0.580 | 4.74±0.02 | 4.465 | 2.35±0.02 | 2.387 | −7.14±0.02 | −6.983 |
| PB$_{11}$ | 12.5 | 0 | 12.5 | 15.89±0.20 | 15.757 | 0.328±0.005 | 0.325 | 0.568±0.005 | 0.557 | 4.13±0.02 | 4.219 | 2.85±0.02 | 2.869 | −4.32±0.05 | −4.603 |
| PB$_{12}$ | 12.5 | 25 | 12.5 | 12.34±0.12 | 12.779 | 0.285±0.018 | 0.292 | 0.573±0.004 | 0.589 | 4.70±0.04 | 4.393 | 2.58±0.04 | 2.621 | −5.89±0.04 | −5.617 |
| PB$_{13}$ | 12.5 | 12.5 | 0 | 14.94±0.16 | 15.035 | 0.318±0.005 | 0.317 | 0.555±0.008 | 0.556 | 4.29±0.04 | 4.255 | 2.71±0.04 | 2.777 | −5.45±0.08 | −5.371 |
| PB$_{14}$ | 12.5 | 12.5 | 25 | 11.45±0.11 | 11.661 | 0.279±0.012 | 0.284 | 0.578±0.001 | 0.582 | 4.70±0.01 | 4.517 | 2.5±0.001 | 2.493 | −6.32±0.01 | −6.409 |
| PB$_{15}$ | 12.5 | 12.5 | 12.5 | 13.32±0.18 | 13.277 | 0.301±0.008 | 0.298 | 0.589±0.003 | 0.575 | 4.33±0.01 | 4.382 | 2.66±0.05 | 2.613 | −5.69±0.03 | −5.718 |
| PB$_{16}$ | 12.5 | 12.5 | 12.5 | 13.27±0.15 | 13.277 | 0.299±0.018 | 0.298 | 0.572±0.002 | 0.575 | 4.35±0.05 | 4.382 | 2.65±0.05 | 2.613 | −5.78±0.02 | −5.718 |
| PB$_{17}$ | 12.5 | 12.5 | 12.5 | 13.01±0.16 | 13.277 | 0.31±0.018 | 0.298 | 0.570±0.005 | 0.575 | 4.35±0.05 | 4.382 | 2.66±0.05 | 2.613 | −5.72±0.05 | −5.718 |
| PB$_{18}$ | 12.5 | 12.5 | 12.5 | 13.89±0.42 | 13.277 | 0.306±0.015 | 0.298 | 0.585±0.001 | 0.575 | 4.33±0.05 | 4.382 | 2.63±0.05 | 2.613 | −5.71±0.01 | −5.718 |
| PB$_{19}$ | 12.5 | 12.5 | 12.5 | 13.56±0.11 | 13.277 | 0.295±0.015 | 0.298 | 0.577±0.002 | 0.575 | 4.50±0.05 | 4.382 | 2.62±0.05 | 2.613 | −5.92±0.02 | −5.718 |
| PB$_{20}$ | 12.5 | 12.5 | 12.5 | 13.23±0.10 | 13.277 | 0.291±0.015 | 0.298 | 0.571±0.005 | 0.575 | 4.10±0.05 | 4.382 | 2.58±0.05 | 2.613 | −5.51±0.05 | −5.718 |

**Table 4. The statistical results using ANOVA for textural and total acceptability confirmed models in burger analysis of turkey meat products.**

| Source | Hardness (N) | | Cohesiveness | | Elasticity (mm) | | Gumminess (N) | | Chewiness (N) | | Adhesiveness (g.s) | |
|---|---|---|---|---|---|---|---|---|---|---|---|---|
| | F-value | p-value | F-value | p-value | F-value | p-value | F-value | p-value | F-value | p-value | F-value | p-value |
| Model | 77.35 | < 0.0001 | 21.62 | < 0.0001 | 9.65 | 0.0007 | 41.02 | < 0.0001 | 20.69 | < 0.0001 | 31.67 | < 0.0001 |
| A-Quinoa | 291.27 | < 0.0001 | 75.28 | < 0.0001 | 21.05 | 0.0010 | 133.65 | < 0.0001 | 79.41 | < 0.0001 | 102.19 | < 0.0001 |
| B-Soybean | 149.12 | < 0.0001 | 49.89 | < 0.0001 | 27.02 | 0.0004 | 68.66 | < 0.0001 | 31.80 | 0.0002 | 63.53 | < 0.0001 |
| C-Amaranth | 191.42 | < 0.0001 | 53.65 | < 0.0001 | 17.94 | 0.0017 | 90.04 | < 0.0001 | 64.85 | < 0.0001 | 66.58 | < 0.0001 |
| AB | 3.43 | 0.0937 | 5.25 | 0.0449 | 0.2175 | 0.6509 | 0.4521 | 0.5166 | 5.33 | 0.0436 | 10.92 | 0.0080 |
| AC | 18.26 | 0.0016 | 2.28 | 0.1617 | 5.44 | 0.0419 | 23.58 | 0.0007 | 0.9216 | 0.3597 | 0.0309 | 0.8640 |
| BC | 19.86 | 0.0012 | 0.1925 | 0.6702 | 1.58 | 0.2378 | 20.77 | 0.0010 | 1.24 | 0.2907 | 0.3572 | 0.5634 |
| A2 | 4.99 | 0.0495 | 3.46 | 0.0925 | 2.75 | 0.1280 | 3.47 | 0.0920 | 0.8740 | 0.3719 | 26.28 | 0.0004 |
| B2 | 18.14 | 0.0017 | 5.61 | 0.0393 | 0.1582 | 0.6992 | 21.34 | 0.0010 | 0.5660 | 0.4692 | 25.14 | 0.0005 |
| C2 | 0.0918 | 0.7681 | 0.2921 | 0.6007 | 1.15 | 0.3094 | 0.5846 | 0.4622 | 0.3438 | 0.5706 | 2.01 | 0.1870 |
| Residual | | | | | | | | | | | | |
| Lack-of-Fit | 2.18 | 0.2063 | 1.16 | 0.4366 | 2.33 | 0.1869 | 3.73 | 0.0874 | 4.51 | 0.0619 | 3.56 | 0.0948 |
| R² | 0.98 | | 0.95 | | 0.96 | | 0.97 | | 0.94 | | 0.96 | |
| Adj R² | 0.97 | | 0.90 | | 0.93 | | 0.94 | | 0.90 | | 0.93 | |

Fig 4e, gumminess samples reduced with the simultaneous addition of these variables. The mutual effect demonstrated for BC, where the B variable kept constant, while C was increased and led to a decline in gumminess (Fig 4f).

Table 3 shows that the lowest chewiness ability was observed in PB10 (1.75 N), while the highest was in PB1 (3.22 N). After removing certain variables, no significant effects found in the model according to the following equation:

Chewiness (N) = 2.29 - 0.38A - 0.24B - 0.34C - 0.11AB

The ANOVA results in Table 4 display that model is highly significant with R² (0.94) and an adjusted form (0.90. < 0.05) detected by A, B and C variables in linear mode and also AB interaction on the chewiness of burgers. The contour diagram for A and B (significant effects $p < 0.05$) observed for variables A, B and C in linear mode, as well as for the interaction between variables A and B on burger chewiness. The contour diagram for variables A and B (Fig 4g) illustrated that chewiness decreased as the values of these enhanced variables.

The maximum and minimum adhesiveness attain in PB$_8$ (7.82 g.s) and PB$_1$ (4.52 g.s), as displays in Table 3. The following equation measured after removing the variables that had no significant effect on the model:

Adhesiveness (g.s) = - 5.72 - 0.64A - 0.50B - 0.51C - 0.23AB - 0.62A2 + 0.60B$^2$

ANOVA results indicated that R² 0.96 and R²$_{adj}$ (0.93) certified a considerable level of the model (Table 4). In burger adhesiveness, A, B, and C variables in linear mode, A and also B for quadratic mode and AB interaction outline a significant effect ($p < 0.05$). The three-dimensional diagram of the mutual effect represents for AB variables in Fig 4h. The adhesiveness value for treated samples declined by elevating variables simultaneously.

The consist of present results, hardness, gumminess, and chewiness for vegetarian hamburgers were reported to be much lower than beef products [5]. The hardness declined by the higher inclusion of pea flour (from 10 to 30%), which interfered with protein-protein interactions [40]. Hardness varied in an extensive range from 400 to 2428 g, elasticity 16.4 to 37.5%, cohesiveness 0.53 to 0.72, and chewiness 276–1530 g had been achieved in vegetable patty samples [24]. The sausages containing more meat require higher grinding power; therefore, the reduction in hardness and cohesiveness replaced with vegetable protein (10–40%) could be because of a weaker myofibril protein network, which declined product resistance to compression [41]. Furthermore, in another study, hardness, chewiness, and gumminess for control were higher, respectively than meat analog samples that illustrated firmer texture because of muscle protein denaturation in the

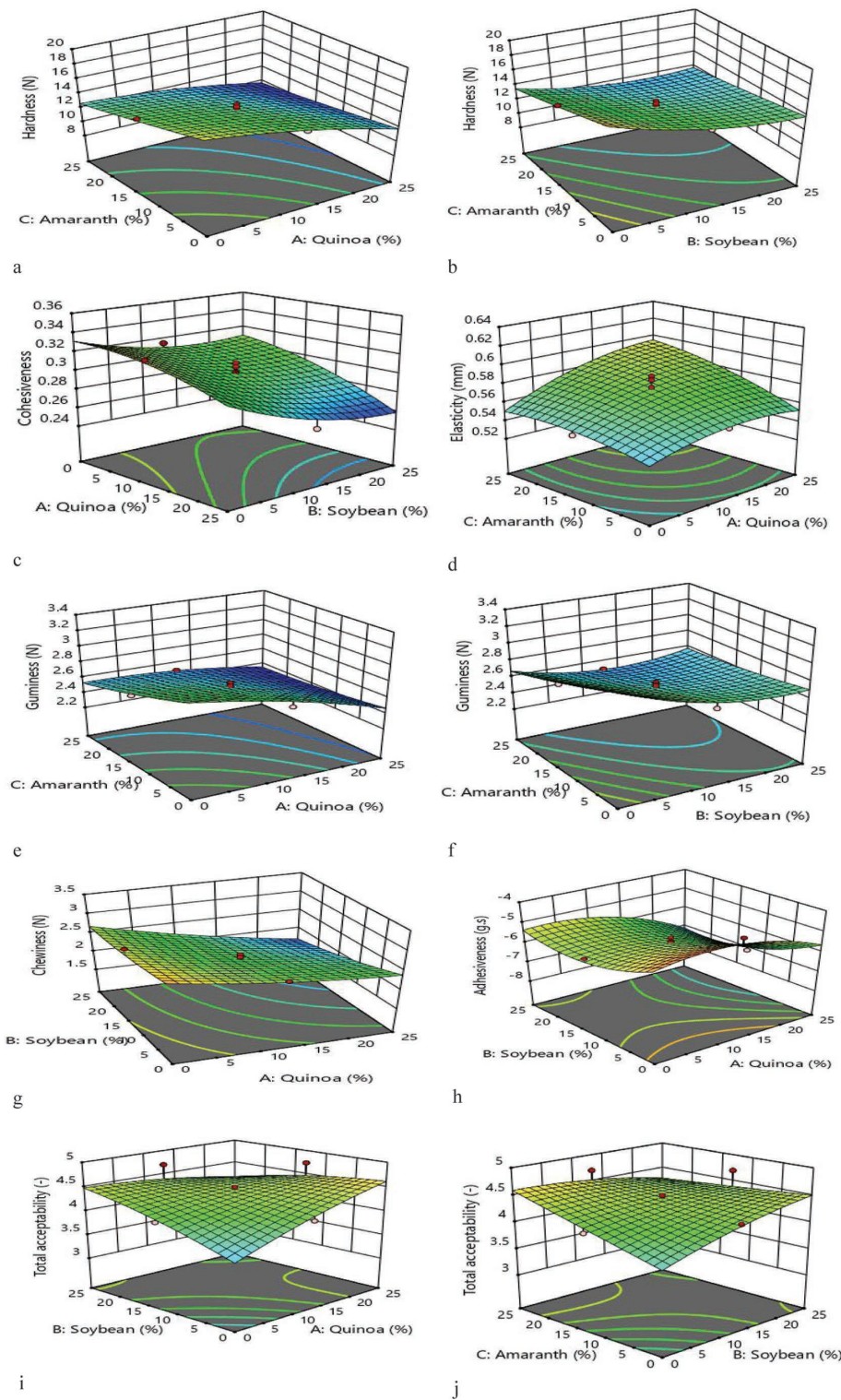

**Fig 4. Three-dimensional (3D) response surface plots indicating the significant (*p*<0.05) interaction effect of factors (A: Quinoa, B: Soybean, C: Amaranth at 0 to 25% concentrations) on the burger.**

system [5]. In a study, chicken burgers with adding amaranth and pumpkin seeds were more efficient compared to others [17]. This increase could be due to the more charged amino acids in soy, and amaranth compared to quinoa, which interacted with lysine, glutamic and aspartic acid of meat myofibrillar proteins and by non-covalent bonds improved hardness, springiness, adhesiveness and gumminess [11].

**Total acceptability of sensory evaluation.** The least overall acceptance obtained in $PB_1$ (3.40) and the most for $PB_6$ (4.95) in **Table 5** and had no significant effect on the model after removing variables based on the below equation:

Total acceptability = 4.38 - 0.31 AB – 0.23 BC

Since the p value calculated for the model was lower than 0.05, which indicated the significance and $R^2$ (0.85) and also $R^2_{adj}$ (0.78) were obtained **Table 5**. The interaction between AB and BC had a significant impact ($p < 0.05$) on the total acceptability of burgers. The overall acceptance increased up to 12.5% protein concentrate in burgers and then decreased with AB and BC variables simultaneously (**Fig 4i**). The overall acceptance trend observed up to 12.5 BC interaction with improving variables, which caused the reduction level (**Fig 4j**).

Turkey meat had a spicy smell, which covered by the replacing plant proteins that did not have a specific smell, and quinoa had a more significant on the higher score compared to soy and amaranth.

In the sensory evaluation of meat nuggets containing quinoa and amaranth flour, they scored high in all cases [32]. For sausage containing quinoa and amaranth flour, more points were observed in sensory evaluation [26]. In another study, all hamburgers with added amaranth or pumpkin seeds rated by participants as having similar or more overall acceptability than the control group [17]. The sensory characteristics were enhanced in tilapia balls, which wheat replaced with barley, quinoa, and amaranth flour [12].

**Optimization of variables based on RSM design.** The optimal formulation identified as containing 25% quinoa, 11.86% soybean and 25% amaranth concentrates. The measured and predicted values for hardness (10.11 ± 0.09 and

**Table 5. Central composite design of texture and total acceptability responses based on independent variables.**

| Run | Independent variables (%) | | | Total acceptability | | Sources | F-value | p-value |
|---|---|---|---|---|---|---|---|---|
| | Quinoa | Soybean | Amaranth | Actual value | Predicted value | | | |
| PB₁ | 0 | 0 | 0 | 3.40 ± 0.01 | 3.105 | **Model** | 3.05 | 0.0484 |
| PB₂ | 25 | 0 | 0 | 4.25 ± 0.03 | 4.463 | **A-Quinoa** | 3.02 | 0.1130 |
| PB₃ | 0 | 25 | 0 | 4.15 ± 0.02 | 4.379 | **B-Soybean** | 0.9629 | 0.3496 |
| PB₄ | 25 | 25 | 0 | 4.60 ± 0.01 | 4.487 | **C-Amaranth** | 2.18 | 0.1703 |
| PB₅ | 0 | 0 | 25 | 4.10 ± 0.02 | 4.267 | **AB** | 9.94 | 0.0103 |
| PB₆ | 25 | 0 | 25 | 4.95 ± 0.01 | 4.775 | **AC** | 4.60 | 0.0577 |
| PB₇ | 0 | 25 | 25 | 4.75 ± 0.02 | 4.591 | **BC** | 5.74 | 0.0376 |
| PB₈ | 25 | 25 | 25 | 3.50 ± 0.01 | 3.849 | **A²** | 0.1759 | 0.6838 |
| PB₉ | 0 | 12.5 | 12.5 | 4.10 ± 0.01 | 4.157 | **B²** | 0.2016 | 0.6630 |
| PB₁₀ | 25 | 12.5 | 12.5 | 4.74 ± 0.02 | 4.465 | **C²** | 0.0006 | 0.9812 |
| PB₁₁ | 12.5 | 0 | 12.5 | 4.13 ± 0.02 | 4.219 | **Residual** | | |
| PB₁₂ | 12.5 | 25 | 12.5 | 4.70 ± 0.04 | 4.393 | **Lack-of-Fit** | 4.77 | 0.0557 |
| PB₁₃ | 12.5 | 12.5 | 0 | 4.29 ± 0.04 | 4.255 | **R²** | 0.85 | 0.0484 |
| PB₁₄ | 12.5 | 12.5 | 25 | 4.70 ± 0.01 | 4.517 | **Adj R²** | 0.78 | 0.1130 |
| PB₁₅ | 12.5 | 12.5 | 12.5 | 4.33 ± 0.01 | 4.382 | | | |
| PB₁₆ | 12.5 | 12.5 | 12.5 | 4.35 ± 0.05 | 4.382 | | | |
| PB₁₇ | 12.5 | 12.5 | 12.5 | 4.35 ± 0.05 | 4.382 | | | |
| PB₁₈ | 12.5 | 12.5 | 12.5 | 4.33 ± 0.05 | 4.382 | | | |
| PB₁₉ | 12.5 | 12.5 | 12.5 | 4.50 ± 0.05 | 4.382 | | | |
| PB₂₀ | 12.5 | 12.5 | 12.5 | 4.10 ± 0.05 | 4.382 | | | |

7.9 N), cohesiveness (0.240±0.001 and 0.260), elasticity (0.610±0.002 and 0.590 mm), gumminess (2.39±0.02 and 2.350), chewiness (1.59±0.04 and 1.640 N), adhesiveness (−7.49±0.03 and −7.62 g.s) and total acceptability (4.69±0.04 and 4.41%) were found for optimal sample, respectively. The values obtained at the optimal points analyzed to validate the results indicating that actual levels closely matched the predicted values confirming the model validity.

The sensory evaluation results particularly the high overall acceptability of PB$_6$ (4.95) reflected the positive influence of physicochemical features. Plant-based samples scored well due to their uniform and appetizing appearances that enhanced by fiber and also ash content, which gave a wholesome look. Neutral flavor of quinoa and soybean ability to mask spicy turkey meat improved smell scores, as plant proteins reduced off-odors [42]. High WHC and ES in plant-based samples created a springy and cohesive texture that reflected in high elasticity scores for PB$_6$, aligning with RSM results showing improved elasticity with quinoa and amaranth. The optimal formulation (25% quinoa, 11.86% soybean and 25% amaranth) balanced texture, flavor and juiciness that driven by high WHC, OHC, EA, ES and low cooking loss. This synergy maximized consumer preference, as seen in PB$_6$ with high score. For instance, plant proteins like soy and pea enhanced WHC and also ES with improving sensory texture in sausages [41]. Similarly, amaranth and pumpkin seeds in chicken burgers improved sensory acceptability by reducing hardness and enhancing juiciness [17], consistent with findings for quinoa and amaranth. The neutral flavor of quinoa complemented sensory profiles by minimizing off-flavors [6], which supported the high smell scores in this study.

**Microstructure of burger samples.** The control microstructure with lacked protein concentrated compared to plant burger with optimal formula in **Fig 5**. The optimal burger sample exhibited smoother, more uniform and compact cavities than the control. The gel structure in the optimal burger was denser, featuring small and homogeneous holes, while control

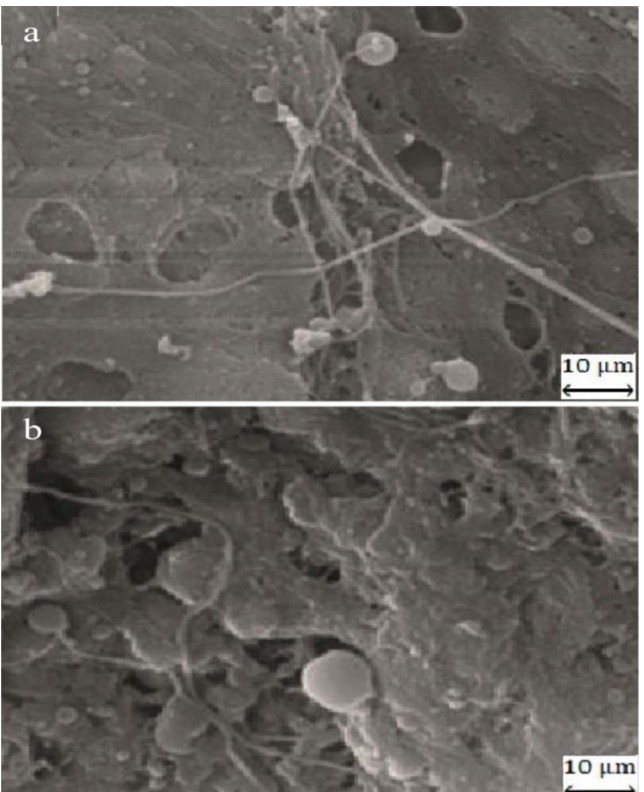

**Fig 5. The microstructure characterization of control and optimal samples in turkey meat burgers.**

displayed a lower uniform distribution of these attributes. These findings suggested that denser gel framework resulted from complex interactions between plant protein concentrate and myofibrillar. Additionally, pore size and arrangement in gel structure influenced water retention and strength for meat products. A dense structure with numerous small holes could minimize water loss and enhance product qualities. The current results indicated that in quinoa sausages, a uniform microstructure of hollow tissue was observed in sample containing 50% protein [15]. In the microstructure investigation of chicken burgers, loose and irregular gel network, and large holes were observed [17]. Similarly, transglutaminase-treated soy and milk protein form a compact structure of gel that could develop the textural functions for chicken sausage; however, when the lipid substitution ratio exceeded 50%, a coarse and cavernous network with large and heterogeneous pores for gel was seen [43]. In another study, the replacement of whey protein with animal fat resulted in a homogeneous microstructure for sausages [44].

## Conclusion

This study produced a low-calorie and nutritionally turkey burger by incorporating 25% quinoa, 11.86% soybean and 25% amaranth protein concentrates that optimized using Taguchi and RSM methods. The formulation excelled in physicochemical attributes (high WHC, OHC, EA, ES and low cooking loss), texture, juiciness and yield. Sensory evaluations favored elasticity and overall acceptability, while SEM confirmed a compact microstructure. Rich in glutamic acid, the burger offered nutritional benefits and reduced health risks from saturated fats. Commercially, it targeted health-conscious and flexitarian markets with potential for other meat products. Suggestions for future development included enhancing flavors with spices to mask off-meat testing lower protein levels (10–15%) for cost-effectiveness, fortification with vitamins or omega-3 in added nutrition, studying shelf life and microbial safety using sustainable and also local protein sources and conducting industrial-scale trials to determine production feasibility. These mentioned steps will support scalable, sustainable and consumer-friendly meat alternatives.

## Supporting information

**S1 File. Taguchi and CCD**.
(ZIP)

## Acknowledgments

We considerably appreciate all participants who assisted in the present research.

## Author contributions

**Conceptualization:** Zeinab Erfanian.

**Data curation:** Zeinab Erfanian.

**Formal analysis:** Zeinab Erfanian.

**Funding acquisition:** Zeinab Erfanian.

**Investigation:** Zeinab Erfanian.

**Methodology:** Zeinab Erfanian.

**Project administration:** Marjan nouri.

**Resources:** Marjan nouri.

**Software:** Zeinab Erfanian.

**Supervision:** Marjan nouri.

**Visualization:** Marjan nouri.

**Writing – original draft:** Zeinab Erfanian.

**Writing – review & editing:** Marjan nouri.

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
