## [Decision Letter · Decision Letter 0]

Dear Dr. nouri,

We look forward to receiving your revised manuscript.

Kind regards,

Karthikeyan Venkatachalam, Ph.D.

Academic Editor

PLOS ONE

Journal Requirements:

Reviewers' comments:

Reviewer's Responses to Questions

**Comments to the Author**

1. Is the manuscript technically sound, and do the data support the conclusions?

Reviewer #1: Yes

Reviewer #2: No

Reviewer #3: Partly

2. Has the statistical analysis been performed appropriately and rigorously?

Reviewer #1: Yes

Reviewer #2: Yes

Reviewer #3: Yes

3. Have the authors made all data underlying the findings in their manuscript fully available?

Reviewer #1: Yes

Reviewer #2: Yes

Reviewer #3: Yes

4. Is the manuscript presented in an intelligible fashion and written in standard English?

Reviewer #1: Yes

Reviewer #2: Yes

Reviewer #3: Yes

Reviewer #1: PLOS ONE

Novel potential of low calorie plant burger: Functional turkey meat formulation

optimized by replacing quinoa, chia, soy, amaranth and peas as vegetable protein and

their influence on texture and sensory traits

General comment: The manuscript in reference compiles the reported “Novel potential of low calorie plant burger: Functional turkey meat formulation optimized by replacing quinoa, chia, soy, amaranth and peas as vegetable protein and their influence on texture and sensory traits”. The manuscript has relevant information and organization that will be interesting for readers. However, some points need to be addressed before further consideration.

1. Abstract: Should be rewritten, and important data was shown.

2. Introduction:

-Should be added more research related to statistics on current global trends in burger and plant-based food consumption.

-What are the highlights of this article? Should add details.

3. Method:

- Should be shown a flow chart as an illustration of Preparation of turkey burger samples.

-Should be added details of equipment in this study, including model, brand, and country of manufacture.

-Sensory evaluation, should be added a detailed description of the criteria for selecting test subjects in accordance with human research ethics standards.

4. Results and discussion:

- Should be explained the meaning of the symbols a, b, c…… in Table 1 and Table 2.

- Should be added more discussion and related research of diagrams obtained in the Taguchi scheme (Figure 2).

- Table 2 and Figure 2, If there is any overlap in the data, the data should be combined or presented as a Table or Figure to reduce confusion and be to the point.

- Should be explained the relationship between physicochemical properties and sensory evaluation.

5. Conclusion:

-Should be rewritten and summarized important points that consistent with the article objectives

-Should be added suggestions, utilization, and guidelines for future development.

6. Other Suggestions:

-Should be checked the reference format according to the journal's requirements.

-Should be updated references.

Reviewer #2: The article needs minor revision.you need use update references from 2021 until 2025. Please revised the article by journal instruction for authors.

Shorter articles with more useful content look better.

Reviewer #3: This study represents a timely and interesting investigation into plant-based protein alternatives in turkey burgers using both Taguchi design and Response Surface Methodology (RSM) to optimize formulations. The topic is relevant to the growing interest in healthier, sustainable meat alternatives. The study provides valuable insights for food science and product development, especially for the formulation of hybrid burgers made from meat and plants. The use of Taguchi and RSM adds robustness to the optimization process. The combination of multiple plant proteins (quinoa, chia, soy, amaranth, and pea) in varying ratios is new in the context of turkey products. The inclusion of physicochemical, textural, sensory, and morphological (SEM) evaluations provides a comprehensive profile of product quality. The manuscript is well organized and thorough, although certain issues of linguistic clarity, data interpretation and scientific rigor should be addressed to meet journal standards

Major Concerns

The manuscript requires language editing. Numerous grammatical errors and awkward phrasings hinder comprehension. For example, "The non-meat proteins apply in food products to improve their yield challenges..." → should be rephrased to "Non-meat proteins are applied in food products to improve yield, nutritional value, and reduce costs." Phrases like “more important micronutrients,” “the minimum saturated fat,” and “a special place in consumer basket” are unclear and need revision.

• The Taguchi design and RSM implementation require more clarity. The orthogonal arrays, levels, and selection rationale are not sufficiently detailed.

• Some statistical analysis details (e.g., model validation, assumption checks, post hoc tests) are missing or unclear.

• Please clarify how replicates were handled in the experiments. For example, were the sensory panels repeated?

• Some conclusions are overstated. For instance, the claim that “produced turkey burgers had the capability of mass production and attracting consumer attention” should be supported with more industrial feasibility data.

• The discussion on amino acid composition needs more biological relevance. Are these differences meaningful in human nutrition? Also, the term “non-substances” for amino acids is confusing.

• There’s limited detail on panelist training, randomization, and blinding. These are crucial to validate sensory data.

• The scoring system seems inconsistent — some terms used (e.g., “poor (3), very poor (2), rejected (1)”) are subjective and may bias results without calibration.

Minor Issues

• References are sometimes outdated; newer literature on plant-based meat analogs (2020–2024) should be cited.

• Units (e.g., g/g, N, %, mm) should be standardized throughout the paper.

• Equation formatting (e.g., Cooking Loss %) needs correction for clarity.

• Avoid repetitive use of phrases like "present research" and “obtained results demonstrated”.

Therefore, the study has merit but requires some language editing, clarification of methods, and improved data interpretation before it can be considered for publication.

**Do you want your identity to be public for this peer review?** For information about this choice, including consent withdrawal, please see our Privacy Policy

Reviewer #1: No

Reviewer #2: No

Reviewer #3: No

---

## [Author Response · Author response to Decision Letter 1]

10 May 2025

Dear Editor

Thank you very much for your valuable comments and all of them were observed. All corrections were done carefully and all changes were identified inside the manuscript point by point with yellow highlighted and the answer of dear referee is expressed in the following table. It should be noted that the lines mentioned by the esteemed judges have been changed due to the addition of some parts to the manuscript, and this change of lines has been identified. Allow me to express my sincerest gratitude for this opportunity you have given me and I am eternally grateful for your tireless efforts and kindness.

Questions Responses

Reviewer #1

1. Abstract: Should be rewritten, and important data was shown. This study explores the creation of a nutritionally and low-calorie turkey burger by integrating plant-based protein concentrates. Turkey burgers formulated with quinoa, chia, soybean, amaranth and pea proteins at inclusion levels of 0 and 30 %. A Taguchi L8 orthogonal array employed to evaluate the effects on water (WHC) and oil (OHC) holding capacities, cooking loss, chemical properties, emulsion activity and stability. Optimal formulations including quinoa, soybean and amaranth at levels of 0, 12.5 and 25 % developed using response surface methodology with a central composite design, emphasizing texture and overall acceptability and also microstructure analyzed through scanning electron microscopy (SEM). The glutamic acid identified as the most abundant amino acid across all protein types. The burgers made entirely of turkey meat displayed the lowest pH (6.13) and protein content (17.36 %). In contrast, the meat free samples exhibited higher moisture, fat, ash and fiber content along with improved WHC and also OHC and reduced cooking loss. Plant protein formulations showed enhanced elasticity and lower levels of hardness, cohesiveness and chewiness compared to the meat-only samples. Sensory evaluations indicated an inclusion preference for protein concentrates with the optimal formulation consisting of 25 % quinoa, 11.86 % soybean, and 25 % amaranth. SEM analysis confirmed the successful integration of vegetable proteins into the burger matrix. These findings highlighted the potential for mass-producing turkey burgers with reduced meat content, enhanced nutritional value, functional and sensory properties.

2. Introduction:

-Should be added more research related to statistics on current global trends in burger and plant-based food consumption. The introduction was modified.

The previous studies conducted on optimization of meat flavoring manufacture for plant-based products using the Taguchi method [19], protein fiber spinning via rheological and physicochemical analyses [20] and also a gluten and soy-free enriched with anthocyanins microcapsules [21]. However, adding a combination of quinoa, chia, amaranth, soybean and pea protein concentrates did not investigate in turkey burger formulation.

-What are the highlights of this article? Should add details. The study aimed to develop a low-calorie and high-nutrition turkey burger by partially replacing meat with plant-based protein concentrates: quinoa, chia, amaranth, soybean and pea. This research contributed to the development of innovative and healthier meat products by integrating a unique combination for plant-based proteins into a turkey burger. The approach enhanced nutritional value and functional properties, which could help reduce health risks associated with conventional meat products.

3. Method:

- Should be shown a flow chart as an illustration of Preparation of turkey burger samples.

-Should be added details of equipment in this study, including model, brand, and country of manufacture. The brand, model and manufacture country of all equipment listed.

-Sensory evaluation, should be added a detailed description of the criteria for selecting test subjects in accordance with human research ethics standards. Sensory evaluation conducted with 40-trained panelists (20-40 years old, equal male and female) in ISO 8589-compliant booths. Panelists underwent three 2 h training sessions to familiarize them with turkey burger sensory attributes (appearance, color, smell and elasticity) and the scoring system. Training included reference samples (commercial turkey burgers and plant-based analogs) to calibrate perceptions of texture, flavor and aroma, ensuring consistent evaluation. Samples fried for 4 min, coded with random 3-digit numbers and presented in a randomized order to prevent sequence bias. Single-blinding ensured panelists were unaware of sample formulations and attributes were scored on a 5-point hedonic scale: 5 (excellent), 4 (good), 3 (acceptable), 2 (poor), 1 (unacceptable). The scale was calibrated during training to minimize subjectivity with “poor” indicating noticeable defects (e.g., off-odor and gritty texture) and “unacceptable” exhibiting rejection due to severe flaws (e.g., rancid flavor and disintegrating texture). Panelists cleaned plates with water and unsalted crackers served between samples to avoid carryover effects [6].

4. Results and discussion:

- Should be explained the meaning of the symbols a, b, c…… in Table 1 and Table 2. Table 1: Letters a-d indicate significant differences between distinct treatments (each row).

Table 2: Letters a-g indicate significant differences between distinct treatments (each row).

- Should be added more discussion and related research of diagrams obtained in the Taguchi scheme (Figure 2). This portion was corrected.

- Table 2 and Figure 2, If there is any overlap in the data, the data should be combined or presented as a Table or Figure to reduce confusion and be to the point. Table 2 has different data than Figure 2.

- Should be explained the relationship between physicochemical properties and sensory evaluation. The sensory evaluation results particularly the high overall acceptability of PB6 (4.95) reflected the positive influence of physicochemical features. Plant-based samples scored well due to their uniform and appetizing appearances that enhanced by fiber and also ash content, which gave a wholesome look. Neutral flavor of quinoa and soybean ability to mask spicy turkey meat improved smell scores, as plant proteins reduced off-odors [42]. High WHC and ES in plant-based samples created a springy and cohesive texture that reflected in high elasticity scores for PB6, aligning with RSM results showing improved elasticity with quinoa and amaranth. The optimal formulation (25 % quinoa, 11.86 % soybean and 25 % amaranth) balanced texture, flavor and juiciness that driven by high WHC, OHC, EA, ES and low cooking loss. This synergy maximized consumer preference, as seen in PB6 with high score. For instance, plant proteins like soy and pea enhanced WHC and also ES with improving sensory texture in sausages [41]. Similarly, amaranth and pumpkin seeds in chicken burgers improved sensory acceptability by reducing hardness and enhancing juiciness [17], consistent with findings for quinoa and amaranth. The neutral flavor of quinoa complemented sensory profiles by minimizing off-flavors [6], which supported the high smell scores in this study.

5. Conclusion:

-Should be rewritten and summarized important points that consistent with the article objectives Conclusion revised.

-Should be added suggestions, utilization, and guidelines for future development. Conclusion revised.

6. Other Suggestions:

-Should be checked the reference format according to the journal's requirements.

-Should be updated references. The references were corrected.

Reviewer #2:

you need use update references from 2021 until 2025.

Please revised the article by journal instruction for authors.

Shorter articles with more useful content look better. These new sources added to the text.

Chilón-Llico R, Siguas-Cruzado L, Apaza-Humerez CR, Morales-García WC, Silva-Paz RJ. Protein quality and sensory perception of hamburgers based on quinoa, lupin and corn. Foods. 2022; 11(21):3405-3425. https://doi.org/10.3390/foods11213405

Sarkisyan V, Bilyalova A, Vorobyeva V, Vorobyeva I, Malinkin A, Zotov V, Kochetkova A. Optimization of the meat flavoring production process for plant-based products using the Taguchi method. Foods. 2025; 14(1):116-130. https://doi.org/10.3390/foods14010116

Sakai K, Okada M, Yamaguchi S. Protein-glutaminase improves water-/oil-holding capacity and beany off-flavor profiles of plant-based meat analogs. PLoS One. 2023; 18(12):0294637. https://doi.org/10.1371/journal.pone.0294637

Szpicer A, Onopiuk A, Barczak M, Kurek M. The optimization of a gluten-free and soy-free plant-based meat analogue recipe enriched with anthocyanins microcapsules. LWT. 2022; 168:113849. https://doi.org/10.1016/j.lwt.2022.113849

The manuscript was adapted to journal format.

Reviewer #3:

This study represents a timely and interesting investigation into plant-based protein alternatives in turkey burgers using both Taguchi design and Response Surface Methodology (RSM) to optimize formulations. The topic is relevant to the growing interest in healthier, sustainable meat alternatives. The study provides valuable insights for food science and product development, especially for the formulation of hybrid burgers made from meat and plants. The use of Taguchi and RSM adds robustness to the optimization process. The combination of multiple plant proteins (quinoa, chia, soy, amaranth, and pea) in varying ratios is new in the context of turkey products. The inclusion of physicochemical, textural, sensory, and morphological (SEM) evaluations provides a comprehensive profile of product quality. The manuscript is well organized and thorough, although certain issues of linguistic clarity, data interpretation and scientific rigor should be addressed to meet journal standards

The items mentioned in the text corrected in all parts of the manuscript.

Major Concerns

The manuscript requires language editing. Numerous grammatical errors and awkward phrasings hinder comprehension. For example, "The non-meat proteins apply in food products to improve their yield challenges..." → should be rephrased to "Non-meat proteins are applied in food products to improve yield, nutritional value, and reduce costs." Phrases like “more important micronutrients,” “the minimum saturated fat,” and “a special place in consumer basket” are unclear and need revision. "The non-meat proteins apply in food products to improve their yield challenges..." deleted.

“More important micronutrients,” was revised.

“The minimum saturated fat,” phrased deleted.

“A special place in consumer basket” phrase deleted.

• The Taguchi design and RSM implementation require more clarity. The orthogonal arrays, levels, and selection rationale are not sufficiently detailed. In the Statistical analysis section, these items were corrected.

Some statistical analysis details (e.g., model validation, assumption checks, and post hoc tests) are missing or unclear. In the Statistical analysis section, these items were corrected.

Please clarify how replicates were handled in the experiments. For example, were the sensory panels repeated? 2.11. Statistical Analysis

2.11.1. Taguchi design

The Taguchi method screened the effects of five plant-based protein concentrates (quinoa, chia, soybean, amaranth and pea) at two levels (0 and 30 %) on physicochemical attributes, WHC, OHC, ES, EA and cooking loss using an L8 orthogonal array (8 runs, SP1 to SP8). The signal-to-noise ratio used "smaller-the-better" for cooking loss and "larger-the-better" in WHC, OHC, EA and ES. Normality (Shapiro-Wilk) and homogeneity of variances (Levene test) verified and also ANOVA assessed significance (p < 0.05) with post hoc test comparing means for Tukey. Quinoa, soybean, and amaranth selected for RSM based on significant effects and also model validation confirmed predicted vs. observed values (< 5 % deviation).

2.11.2. RSM design optimization

RSM with a CCD optimized quinoa, soybean and amaranth (0, 12.5 and 25 %) for texture and sensory acceptability across 20 runs (PB1 to PB20). A second-order polynomial model fitted with normality, homogeneity and independence verified (Shapiro-Wilk, Levene and Durbin-Watson tests). ANOVA confirmed significance (p < 0.05, R² > 0.90) with non-significant lack of fit (p > 0.05). Tukey test compared means and validation that showed < 10 % deviation between predicted and observed responses.

• Some conclusions are overstated. For instance, the claim that “produced turkey burgers had the capability of mass production and attracting consumer attention” should be supported with more industrial feasibility data. This phrase removed.

• The discussion on amino acid composition needs more biological relevance. Are these differences meaningful in human nutrition? Also, the term “non-substances” for amino acids is confusing. The text corrected.

There’s limited detail on panelist training, randomization, and blinding. These are crucial to validate sensory data. Sensory evaluation conducted with 40-trained panelists (20-40 years old, equal male and female) in ISO 8589-compliant booths. Panelists underwent three 2 h training sessions to familiarize them with turkey burger sensory attributes (appearance, color, smell and elasticity) and the scoring system. Training included reference samples (commercial turkey burgers and plant-based analogs) to calibrate perceptions of texture, flavor and aroma, ensuring consistent evaluation. Samples fried for 4 min, coded with random 3-digit numbers and presented in a randomized order to prevent sequence bias. Single-blinding ensured panelists were unaware of sample formulations and attributes were scored on a 5-point hedonic scale: 5 (excellent), 4 (good), 3 (acceptable), 2 (poor), 1 (unacceptable). The scale was calibrated during training to minimize subjectivity with “poor” indicating noticeable defects (e.g., off-odor and gritty texture) and “unacceptable” exhibiting rejection due to severe flaws (e.g., rancid flavor and disintegrating texture). Panelists cleaned plates with water and unsalted crackers served between samples to avoid carryover effects [6].

• The scoring system seems inconsistent — some terms used (e.g., “poor (3), very poor (2), rejected (1)”) are subjective and may bias results without calibration. Sensory evaluation conducted with 40-trained panelists (20-40 years old, equal male and female) in ISO 8589-compliant booths. Panelists underwent three 2 h training sessions to familiarize them with turkey burger sensory attributes (appearance, color, smell and elasticity) and the scoring system. Training included reference samples (commercial turkey burgers and plant-based analogs) to calibrate perceptions of texture, flavor and aroma, ensuring consistent evaluation. Samples fried for 4 min, coded with random 3-digit numbers and presented in a randomized order to prevent sequence bias. Single-blinding ensured panelists were unaware of sample formulations and attributes were scored on a 5-point hedonic scale: 5 (excellent), 4 (good), 3 (acceptable), 2 (poor), 1 (unacceptable). The scale was calibrated during training to minimize subjectivity with “poor” indicating noticeable defects (e.g., off-odor and gritty texture) and “unacceptable” exhibiting rejection due to severe flaws (e.g., rancid flavor and disintegrating texture). Panelists cleaned plates with water and unsalted crackers served between samples to avoid carryover effects [6].

Minor Issues

• References are sometimes outdated; newer literature on plant-based meat analogs (2020–2024) should be cited. These new sources added to the text.

Chilón-Llico R, Siguas-Cruzado L, Apaza-Humerez CR, Morales-García WC, Silva-Paz RJ. Protein quality and sensory perception of hamburgers based on quinoa, lupin and corn. Foods. 2022; 11(21):3405-3425. https://doi.org/10.3390/foods11213405

Sarkisyan V, Bilyalova A, Vorobyeva V, Vorobyeva I, Malinkin A, Zotov V, Kochetkova A. Optimization of the meat flavoring production process for plant-based products using the Taguchi method. Foods. 2025; 14(1):116-130. https://doi.org/10.3390/foods14010116

Sakai K, Okada M, Yamaguchi S. Protein-glutaminase improves water-/oil-holding capacity and beany off-flavor profiles of plant-based meat analogs. PLoS One. 2023; 18(

---

## [Editor Report · Decision Letter 1]

Novel potential of low calorie plant burger: Functional turkey meat formulation optimized by replacing quinoa, chia, soy, amaranth and peas as vegetable protein and their influence on texture and sensory traits

PONE-D-25-11126R1

Dear Dr. nouri,

We’re pleased to inform you that your manuscript has been judged scientifically suitable for publication and will be formally accepted for publication once it meets all outstanding technical requirements.

Kind regards,

Karthikeyan Venkatachalam, Ph.D.

Academic Editor

PLOS ONE

Additional Editor Comments (optional):

This paper can be accepted for publication.
---

## [Editor Report · Acceptance letter]

PONE-D-25-11126R1

PLOS ONE

Dear Dr. nouri,

I'm pleased to inform you that your manuscript has been deemed suitable for publication in PLOS ONE. Congratulations! Your manuscript is now being handed over to our production team.

Kind regards,

on behalf of

Dr. Karthikeyan Venkatachalam

Academic Editor

PLOS ONE